# Systematic Methods for Isolating High Purity Nuclei from Ten Important Plants for Omics Interrogation

**DOI:** 10.3390/cells11233919

**Published:** 2022-12-03

**Authors:** Ming-Chao Yang, Zi-Chen Wu, Liang-Liang Huang, Farhat Abbas, Hui-Cong Wang

**Affiliations:** 1Guangdong Laboratory for Lingnan Modern Agriculture, Guangdong Litchi Engineering Research Center, Key Laboratory of Biology and Genetic Improvement of Horticultural Crops-South China, College of Horticulture, South China Agricultural University, Guangzhou 510642, China; 2Becton Dickinson Medical Devices (Shanghai) Co., Ltd., Guangzhou 510180, China; 3Department of Life Sciences and Technology, Yangtze Normal University, Chongqing 408100, China

**Keywords:** nucleus isolation, plants, cell division cycling, single nucleus RNA sequencing

## Abstract

Recent advances in developmental biology have been made possible by using multi-omic studies at single cell resolution. However, progress in plants has been slowed, owing to the tremendous difficulty in protoplast isolation from most plant tissues and/or oversize protoplasts during flow cytometry purification. Surprisingly, rapid innovations in nucleus research have shed light on plant studies in single cell resolution, which necessitates high quality and efficient nucleus isolation. Herein, we present efficient nuclei isolation protocols from the leaves of ten important plants including *Arabidopsis*, rice, maize, tomato, soybean, banana, grape, citrus, apple, and litchi. We provide a detailed procedure for nucleus isolation, flow cytometry purification, and absolute nucleus number quantification. The nucleus isolation buffer formula of the ten plants tested was optimized, and the results indicated a high nuclei yield. Microscope observations revealed high purity after flow cytometry sorting, and the DNA and RNA quality extract from isolated nuclei were monitored by using the nuclei in cell division cycle and single nucleus RNA sequencing (snRNA-seq) studies, with detailed procedures provided. The findings indicated that nucleus yield and quality meet the requirements of snRNA-seq, cell division cycle, and likely other omic studies. The protocol outlined here makes it feasible to perform plant omic studies at single cell resolution.

## 1. Introduction

A cell is the fundamental unit of life that can vary prodigiously within an organism. Single cell omic analysis has been applied in humans and animal models to uncover gene or protein expression heterogeneity between different cell types within the same organ. Single-cell RNA sequencing (scRNA-seq) enabled the rapid establishment of organogenesis cell atlases in mice and humans [1,2]. However, by the end of the first quarter of 2022, only denumerable articles on single-cell omics involving plants had been published. The reports primarily focus on model plants such as *Arabidopsis* [3,4], rice [5,6], maize [7,8], tomato [9], peanut [10], and populus [11]. Moreover, the tardy progress in plant single cell research, mainly due to the huge difficulty in protoplast isolation from most plant tissues and/or oversize protoplasts during flow cytometry purification.

Recently, studies on mouse visual cortex and kidney revealed that single-nucleus RNA sequencing (SnRNA-seq) provides comparable gene expression quantitation as well as significant advantages over single-cell RNA sequencing (ScRNA-seq) [12,13]. In snRNA-seq of plants, four obvious advantages have been demonstrated, including easier nucleus isolation in contrast to protoplast isolation, less biased cellular coverage, the absence of protoplast isolation-based transcriptional artifacts, and applicability to archived frozen specimens [14]. The nucleus is a complex and highly specialized organelle of the cell that contains the majority of a cell’s genetic information and regulates a large portion of dominant cellular functions [15]. DNA, RNA, and proteins found in nuclei are all subjects of research. High quality nuclear DNA isolation is the prerequisite for nuclear DNA size and ploidy level determination with flow cytometry largely excludes chloroplast and mitochondrial DNA contamination [16]. Nuclear proteins control various cellular functions and regulate complex regulatory networks. Nuclear proteomics, a technique that is heavily reliant on the purity of the nucleus preparation, will aid in the identification of a refined set of proteins that pinpoint certain activities in nuclei, thereby increasing our understanding of cellular processes [17]. SnRNA-seq and single nucleus assay for transposase-accessible chromatin using sequencing (snATAC-seq) performed in the same cell contribute to the definition of gene regulatory programs at epigenomic and transcriptomic levels [18]. Moreover, DNA methylation has been characterized as an available marker to identify intercellular differences accelerating the study of single-cell and single-nucleus DNA Methylation [19,20]. In addition to transcriptional gene silencing via DNA methylation and histone modification, posttranscriptional gene silencing was unearthed in nuclei [21,22]. Hence, isolation of high-quality nuclei is not only essential for high throughput sequencing but for gene expression regulation network research.

Though single-cell omics research is advancing rapidly, the pace on plants is relatively slow due to the complexity of plant cells. The reports of nucleus isolation methods for some plant species such as *Arabidopsis*, tobacco, populus, etc., are either insufficient purity or/and yield to meet the sequencing requirement or use materials limited to tender organs [9,11,23]. For example, the methodology for nuclear isolation from tobacco, potato and apple modified by Sikorskaite et al. cannot meet the quantity and quality requirement for omics sequencing, since it involves performing percoll/sucrose density gradient centrifugation to a purified nucleus, which can only remove obvious impurities, and nucleus isolation buffers require further improvement to achieve higher isolation efficiency [23]. A recent study on model woody populus revealed that nuclei stained with the nuclear dye DAPI do not separate well from unstained nuclei, resulting in loss resolution and thus impurity contamination [11]. Therefore, there is an urgent need to establish a perfect and comprehensive nuclear isolation method. The development of a high efficiency nucleus isolation system for plants is essential for nuclear omic studies, which are powerful tools to systemically reveal the gene regulatory landscape with high resolution and coverage.

Rice and maize are the two most important food crops, with tomato, soybean, apple, grape, banana, citrus, and litchi accounting for more than half of the world’s cash crop. In the current study, we present robust and comprehensive methods for isolating high-purity, high-quality plant nuclei from the aforementioned ten plants including herbaceous, vine and woody plants. Accurate nucleus/cell measurements are important for more comprehensive and rigorous cytological studies, since the precise amount of cell/nucleus is required for in-depth data analysis. For example, compared with the absolute number of CD4+Treg cells in peripheral blood of healthy people, the absolute number of CD4+Treg cells in new-onset rheumatoid arthritis patients was significantly lower [24]. Another important aspect of understanding endosperm development is the study of changes in the number of endosperm nuclei during embryonic development [25]. Though there are methods for cell counting, most of them have limitations since they are mainly based on counting under a microscope, which may have large errors due to cell overlap, and often require a complex pre-treatment before slicing, and cannot be recycled [25,26]. Currently, the absolute cell number counting method using flow cytometry is widely used in clinical research, but it has yet to be used in plants. In this study, we develop for the first time an assay system for absolute cell counting of plants using a flow cytometer, addressing issues such as too small nuclei and the presence of impurities during cell counting. The counted nuclei are ready for genomic, transcriptional and posttranscriptional analysis after integrating highly efficient nucleus isolation with sorting under a sorting flow cytometer using trucount beads as an internal standard. Additionally, facets with references of isolated and purified nuclei used as material in cell division cycle and single nucleus RNA sequencing studies were provided.

## 2. Methods

### 2.1. Reagents

2-(4-Amidinophenyl)-6-indolecarbamidine dihydrochloride (DAPI) (Solarbio, Beijing, China, Cat. No. C0065);2-(N-morpholino) ethanesulfonic acid (MES) (Sigma-Aldrich, St. Louis, MO, USA, Cat. No. M3671);BD Rhapsody™ Cartridge Reagent Kit (BD Bioscience, San Diego, CA, USA, Cat. No. 633731);BD Rhapsody™ cDNA Kit (BD Cat. No. 633773);BD Rhapsody™ WTA Amplification kit (BD Cat. No. 633801);BD Trucount™ absolute count tubes (BD Cat. No. 340334);Dextran T40 (Macklin, Shanghai, China, Cat. No. D806716);Dithiothreitol (DTT) (Genview, El Monte, CA, USA, Cat. No. CD116);DyeCycle Green (Thermo Fisher Scientific, Waltham, MA, USA, Cat. No. V35004);Ethylene diamine tetraacetic acid (EDTA) (Genview Cat. No. LE118);Ethylene glycol tetraacetic acid (EGTA) (Genview Cat. No. LE119);Ficoll 400 (Sigma Cat. No. F4375);Lysis Buffer (BD Cat. No. 650000064);Mannitol (BBI Cat. No. A600335);MgCl_2_ hexahydrate (BBI Cat. No. A601336);Murine RNase inhibitor (Vazyme, Nanjing, China Cat. No. R301-03);Phosphate buffer solution (1× PBS, 0.01 M, pH7.4) (Gibco BRL, Grand Island, NY, USA, Cat. No. 10010072);PI/RNase Staining Buffer (BD Cat. No. 550825);Potassium chloride (KCl) (BBI, Shanghai, China, Cat. No. A501159);Propidium Iodide (PI) (Thermo Fisher Scientific Cat. No. 00699050);Protease inhibitor cocktail (APExBIO, Houston, TX, USA);Proteinase K (NEB, New England Biolabs, Beverly, MA, USA, Cat. No. P81075);Quick RNA isolation Kit (Huayueyang biotech, Beijing, China);RevertAid First strand cDNA Synthesis Kit (Thermo Fisher Scientific Cat. No. 00994100);RiboLock RNase inhibitor (Thermo Fisher Scientific Cat. No. EO0382);Sampling buffer (BD Cat. No. 650000062);Sodium chloride (NaCl) (Genview cat. No. LS014);Spermidine (Sigma cat. No. S0381);Spermine (Sigma cat. No. S3256);Sucrose (Genview cat. No. CS326);Triton X-100 (BBI cat. No. A600198).

### 2.2. Equipment

A 40 µm diameter cell strainer (Biologix, Camarillo, CA, USA, Cat. No. 15-1040);BD FACSMelody™ Flow Cytometer;BD Rhapsody™ Cartridge Kit Rhapsody Cartridge (BD Cat. No. 633733);BD Rhapsody™ Scanner;Centrifuge tubes (BBI Cat. No. F621004, F607888);Chromatography freezer (Panasonic, Japan, MPR-710)Flow tubes (BD Cat. No. 352054);Flow tubes with 35 µm cell strainer (BD Cat. No. 352235);Hemocytometer (INCYTO, Chungnam-do, Korea, Cat. No. DHC-N01-5);Hemocytometer Adapter (BD Cat. No. 633703);Homogenizer (Vortex 3, IKA, Staufen, Germany);Analog Tube Rotator (MX-RL-E) (DLAB, Beijing, China);Swinging rotor centrifuge (Eppendorf 5804R, Hamburg, Germany);ZEISS Axio Imager D2 microscope.

### 2.3. Solution Setup

Nucleus isolation buffer: An overview of the corresponding nucleus isolation buffer for 10 species is presented in Table 1.

Washing and staining buffer (WS): 0.4 U/µL RNase inhibitor, 1× Protease inhibitor cocktail, 0.1 mM spermine and 0.5 mM spermidine in 1× PBS

Collection buffer: 0.2 U/µL RiboLock RNase inhibitor in sampling buffer (BD Cat. No. 650000062).

Washing and staining buffer for cell cycle (WSD): 1× Protease inhibitor cocktail, 0.1 mM spermine and 0.5 mM spermidine in 1× PBS

### 2.4. Plant Material

*Arabidopsis* (*Arabidopsis thaliana*) and tomato (*Solanum tuberosum*) leaves were collected from seedlings grown in a temperature-controlled growth chamber. Leaves and buds of litchi (*Litchi chinensis*), leaves of grape (*Vitis vinifera*), mandarin orange (*Citrus reticulata*), rice (*Oryza sativa*), maize (*Zea mays*), and banana (*Musa nana*) were obtained from the South China Agricultural University’s experimental orchard/field. Young apple (*Malus domesica*) leaves were harvested from five-year potted seedlings, and hydroponic soybean (*Glycine max*) leaves were obtained. After picking, all materials were quickly immersed in liquid nitrogen and stored in a −80 °C freezer. Figure 1 depicts images of all leaf samples.

### 2.5. Nucleus Isolation Procedure

Prepare 4 mL nucleus isolation buffer and 5 mL washing buffer in a tube for each sample. The isolation buffers for each species have been optimized and are listed in Table 1. The solutions should be kept at 4 °C or on ice.

Set the centrifuges to 4 °C.Precool forceps, mortar and pestle (previously sterilized at 121 °C for 20 min) using liquid nitrogen or place them on ice.Add 1 mL of nucleus isolation buffer in the mortar and put it on ice.Transfer 10 mg frozen tissue to the mortar with buffer on ice using the cooled forceps and quickly grind.Transfer the homogenate to a 2 mL tube and wash the mortar with 0.5 mL isolation buffer, then shake it on a rotary mixer (MX-RL-E) in the chromatography freezer for 15 min.Filter the homogenate into a new tube using a 40 µm diameter cell strainer and wash the strainer with 0.5 mL isolation buffer.Centrifuge at 1000× *g* for 5 min at 4 °C.Discard the supernatant and re-extract the residues twice with nucleus isolation buffer, and centrifuge at 1000× *g* for 5 min at 4 °C.Resuspend the white precipitate using 1 mL washing buffer.Transfer the nuclei to a flow tube topped with a blue 35 µm diameter cell strainer and allow the solution to pass through the gravity-driven filter.

### 2.6. DAPI or PI Staining and Microscopy Procedure

11Transfer 100–300 µL solution from step 10 and add more than 3 volumes of 1× DAPI dye solution.12Put it on ice for 5 min in the dark.13Remove the staining solution by centrifuging it at 1000× *g* for 5 min at 4 °C and discard the supernatant.14Wash the precipitate twice with washing buffer and centrifuge it at 1000× *g* for 5 min at 4 °C.15Resuspend the precipitate with 1–3 mL washing buffer in a flow tube.16Examine the nuclei with a ZEISS Axio Imager D2 microscope equipped with an epifluorescence extension and a DAPI filter. Alternatively, propidium iodide (PI) can be used to stain the nuclei. Add 5 μL PI (8 ng/μL) to 100 μL nucleus solution (10^5^–10^8^ nuclei) and keep on ice for 30 min in the dark. The nuclei should then be examined under a microscope or flow cytometer without being washed.

### 2.7. Procedure of Nuclei Purification on Flow Cytometry

17Filter the nuclei staining with DAPI or PI through 35 μm cell strainer into a 5 mL flow tube.18Apply a 100-μm nozzle on flow cytometry.19Using the stain nucleus sample, generate a dot plot of forward scatter (FSC)-area versus side scatter (SSC)-area to determine the size position of nuclei.20Note: Retrieve the nuclei by repeated sorting and observe under a microscope to determine the specific SSC and FSC position of the nuclei.21Generate a dot plot of SSC-height versus SSC-width to select for SSC single nuclei and exclude doublets.22Generate a dot plot of FSC-height versus FSC-width for further selection of FSC singlets and exclusion of doublets.23Use the unstained nuclei as control and generate a contour plot of PI or DAPI signal to identify positive nuclei.24Add 1.5 mL cold collection buffer in a new flow tube and put it in the right position of BD FACSMelody™. Throughout the process, keep the collection buffer cold.25Conduct sorting to collect 200,000–400,000 events (positive nuclei) using a high-purity sorting mode.26After sorting, immediately centrifuge the collected tube at 1000× *g* for 10 min at 4 °C.27Remove the upper supernatant with care, leaving about 300 µL of lower supernatant to precipitate.

### 2.8. Scanning Procedure on the BD Rhapsody™ Scanner

28Stain 500 μL nuclei solution with 2 μL DyeCycle Green and keep it on ice for 5 min in the dark.29Gently pipet 10 uL into the INCYTO disposable hemocytometer.30Insert the hemocytometer into the Hemocytometer Adapter.31Tap Scan in the BD Rhapsody™ Scanner.32Collect and organize the images.

### 2.9. Procedure of Absolute Quantification of Nuclei in Flow Cytometry

33Add 425 μL washing buffer and 50 μL nuclei from step 10 into the BD Trucount tube and vortex for 30 s.34Add 25 μL PI (Thermo Fisher Scientific Cat. No. 00699050) into the tube followed by vortex.35Incubate at room temperature in the dark for 30 min.36Load the tube into flow cytometry.37Generate a dot plot of PerCP-area versus APC-area to select for the position of Trucount beads, and the Trucount bead number can be obtained.38Generate a dot plot of FSC-area versus SSC-area to select the size position of nuclei.39Using the unstained nuclei as a control, generate a contour plot of the PI signal for identifying positive nuclei. The nucleus number of samples can be acquired in this step.40Acquire the sample for further numerical analysis on the flow cytometry.41Calculate the absolute counts of nuclei using the following formula:
absolute nuclear Count = (target population events/bead events collected) × (beads per test from the package insert).

### 2.10. Assaying Cell Cycle Determination Using Flow Cytometry

42Resuspend the white precipitate from step 8 with 1 mL WSD buffer.43Pipette 20 μL nuclei (about 1 × 106 nuclei) to a new flow tube.44Add 500 μL PI/RNase Staining Buffer (BD Cat. No. 550825) to the tube.45Incubate for 15 min at room temperature.46Filter the solution through a 35 μm cell strainer and load the tube into flow cytometry.47Generate a dot plot of PI-height versus PI-width to establish the position of positive nuclei.48Under the positive nuclei, generate a contour plot of PI-height versus PI-area to select for target nuclei and exclude debris.49Generate a histogram plot of PI-area signal to display cells at the G1 and G2 stages.

### 2.11. Nuclei RNA Extraction and Quantitative Real-Time PCR (qRT-PCR) Analysis

Nuclei (2–4 × 10^5^) were sorted into collection buffer, followed by centrifugation at 1000 g for 5 min, and then 550 μL Lysis Buffer with 27.5 μL proteinase K was added to resuspend precipitate, followed by a vortex for 1 min. The total RNA isolation protocol outlined below is based on a Quick RNA isolation Kit. The RevertAid First strand cDNA Synthesis Kit was used to synthesize first strand cDNA from 11 µL total RNA, according to the manufacturer guidelines. Transcript levels of housekeeping genes Actin of the ten plants were determined by qRT-PCR analysis according to Lai et al. [27]. The gene-specific primer pairs are listed in Appendix A.

### 2.12. Library Construction and Sequencing for Whole Transcriptome Analysis (WTA)

The snRNA-Seq libraries were constructed using the BD Rhapsody™ Cartridge Reagent Kit and BD Single cell 3′ whole transcriptome amplification kit following the manufacturer’s instructions.

### 2.13. Data Analysis and Visualization for SnRNA-Seq

The sequencing results were partitioned using the BD Rhapsody WTA pipeline to generate a cell-gene expression matrix file. The reference genome and GTF annotation files to generate reference index is based on the Litchi Genome [28]. Reads were trimmed according to the Rhapsody WTA pipeline and aligned using STAR v.2.5.2b [29]. The matrix was loaded into SeqGeq™ Software v1.7 (FlowJo LLC, Ashland, CA, USA) for further analysis and visualization. The snRNA-seq data were submitted to the NCBI database (PRJNA902319).

### 2.14. Troubleshooting

Frozen tissue quality deteriorates over time due to repeated tissue handling. This always happens after the samples have been handled or removed from the freezer several times. The problem could be solved by putting enough samples for one or two uses in separate tubes when preparing fresh tissue for freezing for the first time.

Plant samples easily clog the nozzle or contaminate the flow cytometry. Plant cells, unlike animal cells, have hard cell walls with debris that is difficult to completely remove. It can only be filtered to ensure a smooth flow. Make sure to filter the sample before loading it and clean the machine with clean solution and water for at least 30 min after the experiment is finished.

Plant samples high in sugar and phenol are easily browned. Due to browning, genetic makeup frequently and easily degrades in plant samples, particularly woody fruit trees. A reducing agent, such as β-mercaptoethanol or dithiothreitol (DTT) could be added as a solution. We recommended DTT, since mercaptoethanol has an unpleasant odor and inhibits browning for a shorter period of time.

Improper handling of single nucleus absolute count can lead to huge errors. BD Trucount™ absolute count tubes contain a lyophilized pellet (beads), so it is important to make sure the nucleus is added to the Trucount beads and to always handle the samples together with the beads. This is one of the most critical factors for avoiding counting errors.

## 3. Results

### 3.1. Optimization of Nucleus Isolation Buffer for Ten Plants

In recent years, developmental biology has made great strides by using single-cell RNA sequencing, and numerous articles have been published in *Nature*, *Science*, *Cell* and other top ranking Journals. Recently, snRNA-seq has been demonstrated to be an alternative single-cell RNA sequencing technology that overcomes the three limitations of single-cell RNA-seq, namely the inability to accurately capture all cell types, the introduction of stress-induced transcriptional artifacts during single-cell dissociation, and the incompatibility with frozen archival material [12,19]. The development of snRNA sequencing technology sheds light on extensive studies in plants at a single cell resolution for overcoming the difficulty of protoplast isolation and the unpredictability of sample availability. However, no method for isolating nuclei suitable for snRNA sequencing has previously been reported for most plants, especially woody fruit trees. The efficiency and quality of nucleus isolation differ among plant species due to their differential cellular structure and density. The development of nucleus isolation method for different plants may greatly benefit their research using nucleus as material.

In the present study, we established a protocol for isolating leaf nuclei from ten different plants, including *Arabidopsis*, rice, maize, soybean, tomato, banana, grape, citrus, apple, and litchi (Figure 1). In order to obtain the optimal nuclear isolation buffer, we used a single control variable method to compare nuclear dissociation under different conditions such as the type and concentration of salt ion and sugar type, pH, Triton, and so on. A variety of nuclei isolation buffer combinations were developed, optimized, and validated. Litchi, citrus, apple, grape and banana are easily brown in alkaline conditions and require much higher DTT levels, whereas in acidic conditions, relatively low DTT levels may prevent browning. Browning is a process that can lead to the irreversible degradation of nucleic acids [30]. Although it can be inhibited under alkaline conditions by adding DTT, excessive DTT may degrade the quality of the extracted nucleus. However, there was hardly brown in tomato, Arabidopsis, rice, maize, and soybean samples, and the nuclei are released efficiently in alkaline conditions. Therefore, fruit crops tend to isolate nuclei in acidic environment due to their high levels of sugar and phenols, whilst tomato, *Arabidopsis*, rice, maize and soybean are more suitable in slight alkaline conditions (Table 1).

In the present study, we found that MgCl_2_ combined with EGTA and the addition of Ficoll and Dextran resulted in a better dissociated effect and cleaner background in rice and *Arabidopsis*. The effects of Ficoll and Dextran on nuclear isolation varied depending on the species studied. The addition of Ficoll and Dextran improves the isolation quality of rice, *Arabidopsis* and soybean, but not for other species. In addition, higher mannitol concentrations resulted in purer nuclei of rice and *Arabidopsis* when compared to other samples. Furthermore, nuclei from tomato, *Arabidopsis*, rice and maize were found to be easily dissociated under a lower concentration of surface active agents, whereas nuclei from litchi, citrus, apple, grape, banana and soybean required higher Triton-X-100 concentration (Table 1). This could be attributed to differences in leaf structure and species age. Leathery leaves may require more Triton.

Following DAPI staining, the nuclei isolated from 10 species were examined under a fluorescence microscope (Figure 2). The presence of numerous stained nuclei demonstrated that the isolation buffer formula listed in Table 1 was more effective for leaf samples from ten different species.

### 3.2. Flow Cytometry for Nucleus Purification

Even after several washes with a high-sugar buffer, significant amounts of cellular debris and impurities were found in the isolated nuclei, introducing immeasurable interference in further study (Figure 2); therefore, it is indispensable to purify the isolated nuclei. The procedure for flow cytometric nuclei purification is described in detail in the method section. Nuclei stained with PI and then filtered by SSC and FSC showed an obvious PI signal under flow cytometry (Figure 3a–d), suggesting that gated nuclei are superior.

The nuclei from ten crops after being sorted under flow cytometry were monitored by the BD Rhapsody™ Scanner. The images show that the sorted nuclei are of exceptionally high purity and quality, with almost no impurities (Figure 3e and Appendix A). These results declared the high-efficiency nucleus isolation and purification methods for the ten plant species.

### 3.3. The Absolute Nucleus Count of Ten Plant Species

Currently, nuclei are often counted in the blood counting chamber; however, nuclei frequently fail to appear on the same imaging surface under a microscope, since nuclei are much smaller than cells, resulting in underestimation. The BD Biosciences single-platform system (San Jose, Calif.), a simple but precise method for calculating cell numbers in clinical trials, meets the need for accurate and reproducible monitoring of nuclei, especially in rare samples [31,32]. However, this absolute counting method has primarily been used for clinical cell counting rather than nuclei and has yet to be applied to plants. The use of a Trucount tube containing a known number of fluorescent beads is critical to this method. Following the counts of Trucount beads and sample nuclei employing flow cytometry, the absolute nuclei number can be determined using the formula listed in the method section (Table 2). A logical diagram for determining nuclei of ten crops in the flow cytometry is shown in Figure 4a–c. To verify the reliability of the absolute counting method, we determined the nuclei number in 500 μL nuclei solutions with different volumes of nuclei from step 10, and obtained a linear regression curve with a correlation coefficient of 0.9905 (Figure 4d).

In the current study, 10 mg of leaves from ten plants were taken to determine the number of nuclei using flow cytometry. Interestingly, banana, grape and litchi, which use young leaf material, displayed a much higher nuclear yield than the rest of seven plants, which used mature leaf material (Table 2). Among the seven mature leaf samples, the nuclear yield showed a significant positive relationship with dry matter, with *Arabidopsis* having the lowest, while citrus and soybean had much higher (Figure 4e). Taken together, the nucleus yield of the same amount of fresh plant samples might depend on the tissue development stage and dry matter ratio.

### 3.4. Nuclear RNA Quality Monitoring and Application in snRNA-Seq

To preliminarily determine the quality of the sorted nuclei, we collected 200,000 to 400,000 nuclei for qRT-PCR analysis and assessed the expressions of housekeeping genes. According to the amplification curve, the Ct value was around 29, since the transcription level of nuclei is typically lower when compared to whole cells (Figure 5a). Melt curve and melt peak of the ten plants revealed that the melt peak occurs around 80, with only one specific peak (Figure 5b,c). These results indicate that purified nuclei can meet the requirement of RNA extraction and qRT-PCR.

To further evaluate the nuclear RNA quality and demonstrate that the sorted nuclei meet the requirements for sophisticated molecular research, such as snRNA-seq and spatial transcriptome, a whole transcriptome analysis (WTA) library from litchi vegetative bud was constructed based on the BD Rhapsody™ platform and sequenced on Illumina NovaSeq sequencers (Figure 6a). Since nuclei from buds are more difficult to isolate than nuclei from leaves, this was chosen as a material to depict the feasibility of the developed nucleus isolation method. The library peak and gel image showed that the library fragment was primarily concentrated in around 600 bp, with single peak type and no miscellaneous peak, implying that the isolated nuclei were of high quality for single nucleus library construction (Appendix A).

In the current study, 15,597 nuclei were captured, and 27,452 genes were identified after quality control and filtration on SeqGeq. The highly dispersed genes were selected for the t-distribution stochastic neighborhood embedding (t-SNE) analysis, followed by a K-means clustering analysis to yield 16 cell clusters (Appendix A). A heatmap depicted the expression pattern of nine marker genes among the 16 identified clusters and the clusters were subsequently defined and annotated according to specifically expressed marker genes with distinct molecular functions (Figure 6b,c and Appendix A). The mesophyll cell (MC) population was composed of cluster 9, 10, 11 and 14, where photosynthesis related genes LOX2.1 [33,34] and RBCS1 were predominantly expressed [35]. The epidermis cell (EC) population consisted of cluster 4 and 5, in which specifically expressed epidermal-specific genes, such as DCR [3] and FDH, were located [36]. Homeobox-leucine zipper protein HAT5 [37] was predominantly expressed in cluster 16, which functioned as a guard cell (GC). Moreover, G2_M proliferating, meristem cells (MeC), G1_S proliferating cells (G1_S), proliferating mesophyll cells (expressed both CDKB2-2 and RBCS1, PMC) and epidermis plus mesophyll cells (EMC) were also defined [3,34,38]. Cluster 13 was not defined (undefined cells, UC) for any expression of corresponding characteristic marker gene. The high-resolution cell population obtained from snRNA-seq demonstrated the efficacy of our method for obtaining nuclear RNA quality. Together, the nuclear isolation methods mentioned above can be applied to further transcriptional studies.

### 3.5. Nuclear DNA Quality Monitoring

The cell division cycle status of litchi nuclear DNA was determined using flow cytometry to assess its quality. In contrast to RNA degradation inhibition in applying the isolated nucleus at transcriptional level, it is critical to remove RNA thoroughly during application at the genomic level because the inclusion of RNA prompts background contamination. The isolated nuclei can be used to identify different cell division cycle cells, as shown in Figure 7. Cells in the G1 and G2 phases can be clearly identified. These findings suggested that the nucleus isolation methods presented in this study completely met the standards of deep genomic research.

## 4. Conclusions

Single-nucleus RNA and ATAC sequencing, single-nucleus DNA methylation, and the nuclear proteome have become extremely relevant in basic and applied research. We optimized efficient nucleus isolation protocols for *Arabidopsis*, rice, maize, tomato, soybean, banana, grape, citrus, apple, and litchi leaves to improve single cell omic studies in plants. The optimized method is suitable for samples with different ripening degrees, even mature woody fruit tree leaves.

For the first time, we developed an absolute plant cell number counting method by combining flow cytometry with a Trucount tube. This might benefit the studies of the relationship between cell or nuclear numbers and plant development. The isolated nuclei were further purified by sorting under flow cytometry. Furthermore, the quality of isolated nuclei was evaluated using a microscope and qRT-PCR, and an example of snRNA-seq using the nuclei isolated from litchi vegetative bud was presented. When DNA size and ploidy level are determined using our isolated nuclei, chloroplast and mitochondrial DNA contamination can be completely avoided. Additionally, the cell division cycle study of litchi leaves was presented as an example using a nuclear sample prepared according to our protocol.

Taken together, the findings suggest that nucleus quality and yield fulfill the standards of the snRNA-seq, cell division cycle and might also fulfill those of other deep transcriptional and genomic studies.

## Figures and Tables

**Figure 1 cells-11-03919-f001:**
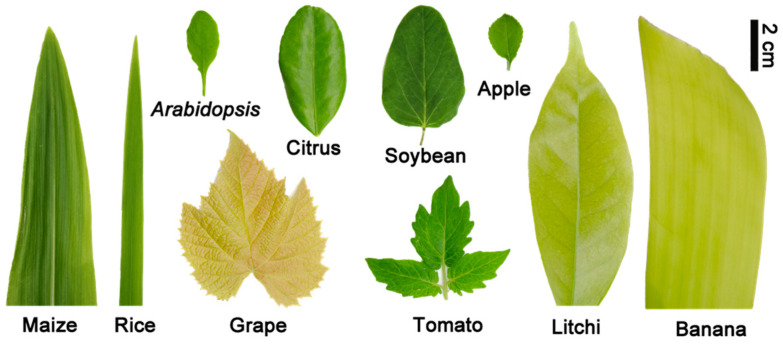
The images of sampling leaves from maize, rice, tomato, arabidopsis, soybean, banana, grape, litchi, citrus, and apple. Scale bars = 2 cm.

**Figure 2 cells-11-03919-f002:**
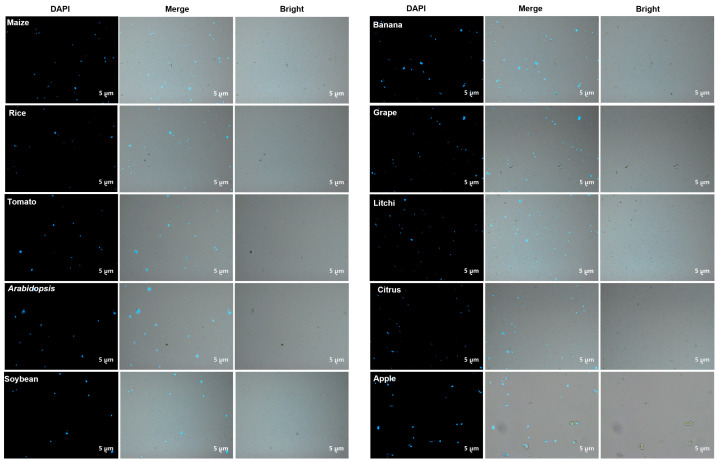
Isolated nuclei of the ten plants before purification observed using a ZEISS Axio Imager D2 microscope. Scale bars = 5 μm.

**Figure 3 cells-11-03919-f003:**
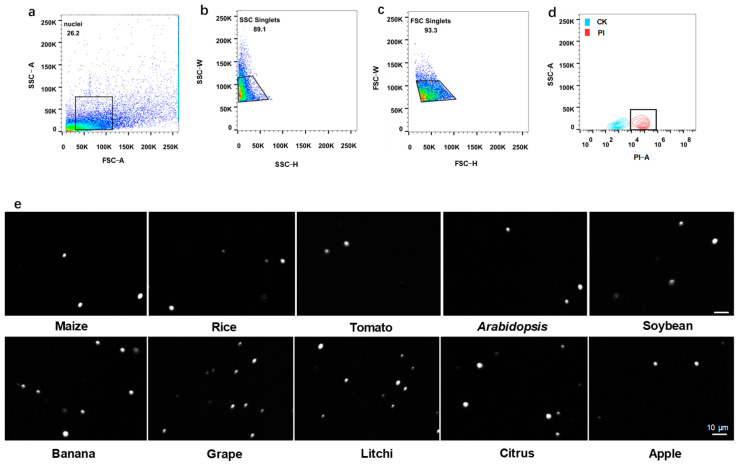
Logic diagram of nuclear quality control using flow cytometry and images of sorted nuclei. (**a**) Dot plot of size position of nuclei selected by FSC versus SSC. (**b**). SSC single nuclei collected by SSC-height versus SSC-width. b plot produce based on the rectangular part of a plot. (**c**) FSC single nuclei collected by FSC-height versus FSC-width. Plot c produce based on the trapezoidal section of b plot. (**d**) Contour plot of PI positive nuclei (red) and control nuclei (blue). Plot d produce based on the trapezoidal section of plot c. (**e**) Sorted nuclei observed under BD Rhapsody™ Scanner from maize, rice, tomato, arabidopsis, soybean, banana, grape, litchi, citrus, and apple.

**Figure 4 cells-11-03919-f004:**
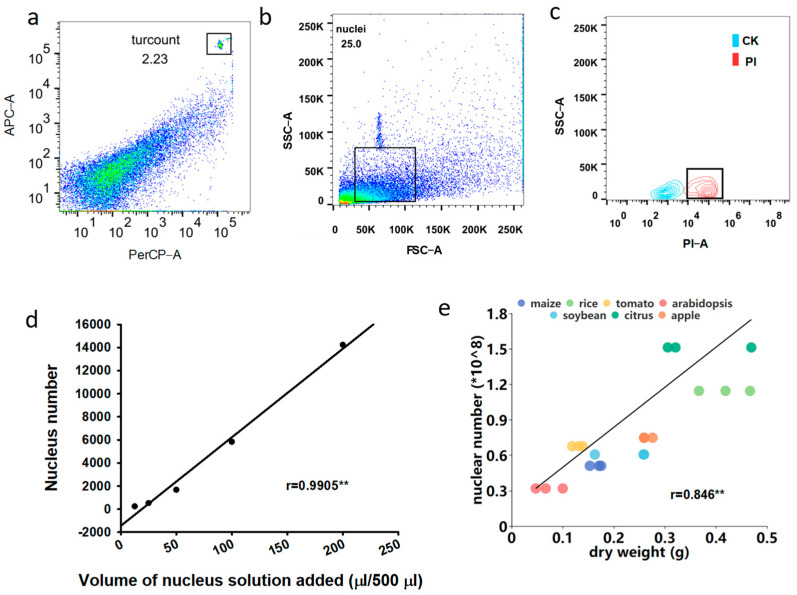
Nuclei absolute quantitative logic using flow cytometry. (**a**) Dot plot of trucount beads selected by PerCP-area versus APC-area. (**b**) Dot plot of nuclear position selected by FSC versus SSC. (**c**) Contour plot of PI positive nuclei (red) and control nuclei (blue). Plot c produce based on the rectangular part of plot b. (**d**) Linear regression curve of nuclei solutions. ** indicate significant correlation at *p* < 0.01. (**e**) The absolute counts of nuclei from leaf sample of maize, rice, soybean, arabidopsis, tomato, citrus and apple were positively correlated with the dry weight.

**Figure 5 cells-11-03919-f005:**
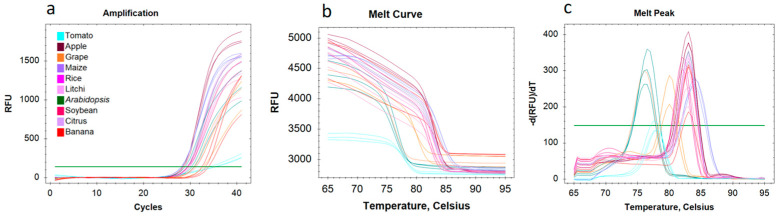
Nuclear RNA quality monitoring. (**a**) Actin amplification curve of the nuclei RNA of ten plants. (**b**) Melt curve of the nuclei of ten plants. (**c**) Melt peak of the nuclei of ten plants. Each color represents a different plant species.

**Figure 6 cells-11-03919-f006:**
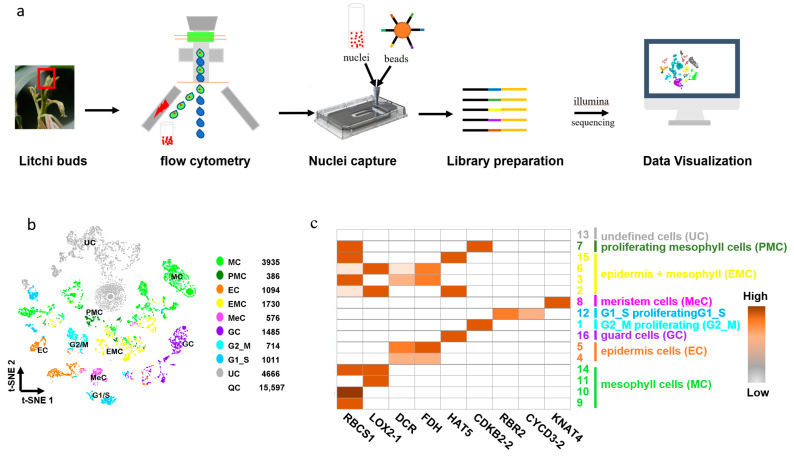
An example of snRNA sequencing using litchi bud nuclei. (**a**) Process of BD Rhapsody™ Single-Cell Analysis. (**b**) t-SNE plot showing 9 cell populations from the litchi bud. Each dot denotes a single cell. Colors denote corresponding cell clusters. (**c**) Expression patterns of representative cluster-specific marker genes. KNAT4: Homebox protein knotted-1-like 4; CYCD1-4: Cyclind3-2; CDKB2-2: Cyclin-dependent kinase B2-2; RBR2: Retinoblastoma-related protein2; LOX2.1: Lipoxygenase 2.1; RBCS1: Rubisco bisposphate carboxylase small subunit 1; HAT5: homeobox-leucine zipper protein HAT5; FDH: Fiddlehead; DCR: Defective in cuticular ridges.

**Figure 7 cells-11-03919-f007:**
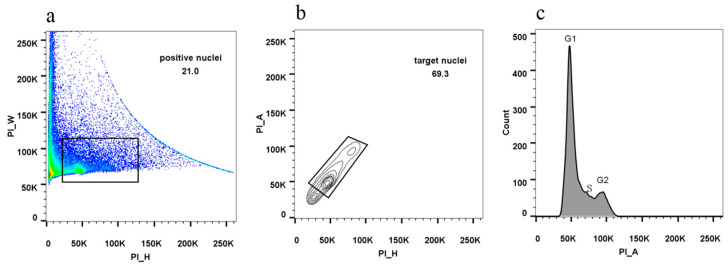
An example of cell cycle study using litchi bud nuclei. (**a**) Dot plot of positive nuclei selected by PI-height versus PI-width. (**b**) Contour plot of nuclei from positive nuclei selected by PI-height versus PI-area. b plot produce based on the positive nuclei (rectangular part of a plot). (**c**) Histogram plot of different cell division cycles collected by PI-area. c plot produces from the target nuclei (rectangular part of b plot).

**Table 1 cells-11-03919-t001:** An overview of the corresponding nuclei isolation buffer for the leaf from 10 plants.

	Maize	Rice	Tomato	*Arabidopsis*	Soybean	Banana	Grape	Litchi	Citrus	Apple
MES-NaOH (10 mM)	7.4	7.4	7.4	7.4	7.4	5.7	5.7	5.7	5.7	5.7
NaCl (mM)	10	-	10	-	10	10	10	10	10	10
KCl (mM)	10	-	10	-	10	10	10	10	10	10
MgCl_2_ (mM)	-	10	-	10	-	-	-	-	-	-
EGTA (mM)	2	2	2	2	2	-	-	-	-	-
EDTA (mM)	-	-	-	-	-	2.5	2.5	2.5	2.5	2.5
Mannitol (mM)	250	400	250	400	250	250	250	250	250	250
Ficoll 400 (%)	-	2.5	-	2.5	0.5	-	-	-	0.5	-
Dextran T40 (%)	-	5	-	5	1	-	-	-	1	-
Spermine (mM)	0.1	0.1	0.1	0.1	0.1	0.1	0.1	0.1	0.1	0.1
Spermidine (mM)	0.5	0.5	0.5	0.5	0.5	0.5	0.5	0.5	0.5	0.5
Cocktail (%)	0.1	0.1	0.1	0.1	0.1	0.1	0.1	0.1	0.1	0.1
RNase inhibitor (U/μL)	0.2	0.2	0.2	0.2	0.2	0.2	0.2	0.2	0.2	0.2
DTT (uM)	1	1	-	-	1	500	500	500	500	500
Triton-100 (%)	0.8	0.8	0.5	0.5	2	1	1	1	2	2

**Table 2 cells-11-03919-t002:** Detailed statistical data of trucount beads, nuclei, dry weight for the leaves from ten plants.

Items	Maize	Rice	Tomato	*Arabidopsis*	Soybean	Banana	Grape	Litchi	Citrus	Apple
Sample amount (FW mg)	10	10	10	10	10	10	10	10	10	10
Dry matter ratio (%)	16.7 ± 0.7	41.8 ± 2.9	13.0 ± 0.6	7.1 ± 1.5	22.7 ± 3.2	11.0 ± 0.1	30.7 ± 0.5	19.0 ± 2.1	36.5 ± 5.2	26.5 ± 0.6
Sample nuclei (No.)	282.5 ± 11.8	119 ± 7.8	170 ± 0.6	644 ± 2.3	558 ± 11.8	1233 ± 157.3	506 ± 3.2	1457 ± 32.9	581 ± 6.4	983 ± 10.8
Trucount beads (No.)	524 ± 2.6	99 ± 4.3	236 ± 3.2	1883 ± 2.3	897 ± 18.5	296 ± 7.8	203 ± 10.1	400 ± 5.8	361 ± 6.6	1340 ± 37.2
Nucleus tested (number ∗ 10^8^ g^−1^ DW)	3.03 ± 0.14	2.73 ± 0.30	5.21 ± 0.05	4.52 ± 0.02	2.57 ± 0.01	35.3 ± 0.36	7.66 ± 0.33	18.0 ± 0.67	4.14 ± 0.03	2.81 ± 0.08

Note: Nucleus No. tested = 46,900 × sample nuclei/Trucount beads × 20/0.01/dry matter ratio × 100 (No. per g dry weight). 46,900 is the known number of Trucount beads per tube and 20 is the dilution times.

## Data Availability

The snRNA-seq data have been deposited in NCBI database (SRA: PRJNA902319).

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
