# Peer review of "Systematic Methods for Isolating High Purity Nuclei from Ten Important Plants for Omics Interrogation"

_cells, 2022, doi:10.3390/cells11233919_

Round 1

Reviewer 1 Report

In this manuscript, Yang et al., reported protocols of nuclei isolation for 10 plants. As the authors mentioned in the manuscript, single-cell analyses have been known as an invention that provides great improvement in spatial resolution and helps to deepen understanding of cellular characteristics, especially in the field of developmental biology. Simultaneously, there has been common recognition that single-cell isolation is a problematic step to perform such studies. The publication of concrete protocol for various plant species will support expanding the single-cell analyses as well as help to examine methods for previously unanalyzed plant materials. 

I think the manuscript will be of interest to readers of Cells, especially those who are interested in plant developmental biology and genetics. Accordingly, I think the manuscript seems to be suitable for publication from the journal, however, at this time, the manuscript submitted needs to be improved regarding the points listed below.

L126

Panasonic”n” MPR-710 : typo?

Plant material or nucleus isolation procedure section,

The authors are preferred to add detailed information for the storage of plant samples. Although leaf samples are sought to be frozen before subjecting the procedure according to the description, a clear description will be helpful.

L156

Is any special requirement for mortar and pestle?

L161

MX-RL-E (DLAB, Beijing): what kind of equipment?

For cell strainer and flow tube,

There are two types of cell strainer, e.g. 40 um (L162) and 35 um (L168) in the protocol; however, only 40 um one was introduced in the Equipment section. The 35 um one is from the Flow tube 352235? I think it is better to introduce these materials (cell strainer, flow tubes) separately to help readers.

L171

step 10?

L186

Which cell strainer is used for this step?

L230

white precipitate : Step8?

L232

PI/RNase Staining buffer: same one with step34? Please use consistent description.

L234

35 um cell strainer?

L236-238

(Figure5a/b/c) : Figure7?

Figure2, 3e

The overall quality of the micrographs is too low, and the sizes are too small. I think the authors are preferred to provide magnified images to help readability.

Table 1

T”ur”count: typo?

snRNA-seq data must be deposited in database, e.g. NCBI GEO.

Figure5

Figure legend is insufficient. Colors and corresponding plant samples must be explained.

Figure6c

Color gauge for the heatmap must be added.

Author Response

Many thanks for the thoughtful comments on our paper. We have made significant changes in response to the reviewers' comments. In order to ensure that our manuscript is error-free in every way, we have thoroughly proofread it. The following are our responses to the reviewer's questions and suggestions:

  1. On L167-168, detailed instructions for storing plant samples have been added.
  2. Mortar and pestle must be sterilized at 121 °C for 20 minutes, as specified on L175.
  3. The text has been updated with information on how to use which types of cell strainers (35 μm and 40 μm).
  4. Spelling and marking errors in the article have been corrected as per the reviewer’s suggestion.
  5. PI/RNase staining buffer in step 44 and the PI in 34 are different. They are listed separately on L237 and L255.
  6. Thank you for the suggestion to improve the figures. Figures 2 and 3e have been replaced with magnified images. The figures are much clearer now.
  7. Figure 5 depicted the colors and corresponding plant samples, and Figure 6c added a color gauge for the heatmap.
  8. The snRNA-seq data was submitted to NCBI (PRJNA902319) and cited in the text.(L283).

Reviewer 2 Report

The submitted manuscript reports a systematic methods for isolating high purity nuclei from ten important plants for omics interrogation. Utilizing the optimized nucleus isolation buffer, high nuclei yields from ten different plants could be achieved. Moreover, through an absolute plant cell number counting method, the relationship between nuclei numbers and plant can be identified.

The study shows the detailed procedure for nuclei purification. However, given the fact that previous publications (e.g. PLoS ONE 16(5): e0251149 and Plant Methods 2013, 9:31) have already demonstrated using similar purification conditions and procedures, the novelty that the authors claim is limited. On the other hand, the author emphasized the absolute plant cell number counting method that they developed is unprecedented, but I found it would be difficult to be a selling point. Since this method is widely used in other topics such as J. Anat. (2016) 229, pp406-415.

Overall, I believe the current manuscript has insufficient fundamental significance for Cells and leaves readers with no new method to learn. It would be more appropriate for publication in a more specialized journal, or it would need a major revision to add more detail. I believed with additional information added as suggested below, it would be an educational story and interesting to the readers.

Suggestions and comments for revision:

1.     Put these 2 papers (PLoS ONE 16(5): e0251149 and Plant Methods 2013, 9:31) into the introduction and talks about the difference between the tcurrent study procedure and these currently developed protocols.

PLoS ONE 16(5): e0251149 (A robust method of nuclei isolation for single-cell RNA sequencing of solid tissues from the plant genus Populus)

Plant Methods 2013, 9:31(Protocol: Optimised methodology for isolation of nuclei from leaves of species in the Solanaceae and Rosaceae families)

2.     The authors summarized the optimized isolation buffer in table 1 but leave the reader with no explanation or comparison of how different compared to different buffers that they have tried. It would be very helpful if the author can add more supporting information regarding the outcome of different isolation buffers and provide more details so that the reader could gain more information about the merit of the current procedure. For example, why rice needs NaCl and KCl instead of MgCl2? Have the authors tried the buffer that is used to isolate maize on rice and what is the outcome?

3.     In table 1, page 9, MgCl should be MgCl2

Round 2

Reviewer 1 Report

The authors have revised the manuscript responding to the concerns that are raised by the reviewers. I think the manuscript seems to be suitable for publication from the journal.

Reviewer 2 Report

The current submission successfully resolved all the issues that I mentioned earlier and it’s well-written and educational. In my opinion, it is suitable to be published on Cells and it would be interesting to the readers. Congratulations to the authors, I am happy to recommend publication.